# Transcriptional repression of *reaper* by Stand still ensures female germline development in *Drosophila*

Masaya Matsui[1], Shinichi Kawaguchi[1], Toshie Kai[1,2]*

1 Graduate School of Frontier Biosciences, The University of Osaka, Osaka, Japan, 2 Graduate School of Biostudies, Kyoto University, Kyoto, Japan

* kai.toshie.2w@kyoto-u.ac.jp

## Abstract

Apoptosis plays a central role in shaping tissues and preserving cellular integrity across developmental stages. In the germline, its precise regulation is critical to ensure both the elimination of aberrant cells and the maintenance of reproductive capacity. However, the molecular mechanisms that control apoptotic susceptibility in germline cells remain poorly defined. Here, we identify *stand still* (*stil*) as a female germline-specific regulator of apoptosis in *Drosophila*. Loss of *stil* leads to near-complete depletion of germline cells at the time of eclosion, associated with upregulation of the pro-apoptotic gene *reaper* (*rpr*) and activation of caspase-dependent cell death. Reporter assays in S2 cells show that Stil directly represses *rpr* transcription through its N-terminal BED-type zinc finger domain. The Dietera-restricted conservation of *stil* and *rpr* is consistent with a functional association. Despite the absence of *stil*, undifferentiated germline cells remain resistant to apoptosis. Analysis of publicly available chromatin data reveals that the *rpr* locus in these cells resides in a closed, H3K9me3-enriched chromatin state, suggesting a Stil-independent mode of transcriptional silencing. Together, our findings uncover two distinct mechanisms that protects the female germline from *rpr*-dependent apoptosis: Stil-mediated transcriptional repression that operates in both undifferentiated and differentiated germline cells, and an additional chromatin-based silencing mechanism that functions specifically in undifferentiated cells. This work provides new insights into the interplay between transcriptional and chromatin-based regulations that maintain germline cell identity and survival.

## Author summary

Apoptosis eliminates unnecessary or abnormal cells, yet its inappropriate activation in germline cells would be highly detrimental to reproduction. A finely tuned mechanism that permits cell death only when necessary while preventing its

**Data availability statement:** NGS data set have been deposited to the DNA Data Bank of Japan (DDBJ). BioProject Accession: PRJDB20709 and PRJDB20722. All fly strains generated for this study are available upon request.

**Funding:** This work was supported by the Japan Science and Technology Agency (JPMJSP2138 to MM), Grant-in-Aid for Transformative Research Areas (A) (21H05275 to TK), the Japan Society for the Promotion of Science (T25K022050 to TK), and the Naito Foundation (J161503009 to TK). The funders had no role in study design, data collection and analysis, decision to publish, or preparation of the manuscript.

harmful activation is indispensable for maintaining fertility and species continuity. In this study, we identified *stand still* (*stil*) as a factor that prevents inappropriate activation of the pro-apoptotic gene *reaper* (*rpr*) in the female germline of *Drosophila*. Loss of *stil* resulted in near-complete depletion of ovarian germline cells at eclosion, accompanied by elevated *rpr* expression and activation of caspase-dependent apoptosis. Furthermore, we demonstrate that Stil directly represses *rpr* transcription through its N-terminal zinc finger domain. In addition, undifferentiated germline cells, including germline stem cells, exhibit a closed chromatin state at the *rpr* locus, suggesting that *rpr* is intrinsically less accessible for transcription in these cells. Together, this two-layered mechanism suggests that susceptibility to cell death is tuned according to developmental stage, providing a conceptual framework for understanding reproductive and developmental abnormalities as well as diseases associated with dysregulated cell death.

## Introduction

Apoptosis, a form of programmed cell death (PCD), is a highly regulated biological process essential for animal development and tissue homeostasis. *Drosophila* is an excellent model organism for studying apoptotic cell death, due to its powerful genetic and molecular tools. In *Drosophila*, apoptosis is often initiated by the expression of IAP (inhibitor of apoptosis) antagonists, including members of the RHG gene family: *reaper* (*rpr*), *hid* (*head involution defective*), *grim*, and *sickle* (*skl*) [1–4]. Among these, the pro-apoptotic gene *rpr* plays a central role in PCD and stress-induced apoptosis [5]. The *Drosophila* IAP protein DIAP1 is degraded via auto-ubiquitination, triggered by the physical interaction of Rpr through its conserved IAP-binding motif (IBM), leading to caspase activation and apoptotic cell death [6,7]. *rpr* expression is regulated by transcription factors and histone modifications, such as H3K27 and H3K9 trimethylation, in response to developmental and environmental signals [8–12].

For reproductive success, apoptosis is meticulously regulated to control cell numbers and maintain germline cell quality [13–16]. *Drosophila* adult female has a pair of ovaries, each containing 16–20 ovarioles. At the apical end of each ovariole lies the germarium, which houses germline stem cells (GSCs) that divide asymmetrically to produce a self-renewing GSC and a cystoblast. Cystoblast undergoes four mitotic divisions to form a 16-cell cyst interconnected by ring canals. Of these 16 cells, one differentiates into the oocyte and enters meiosis, while the remaining 15 become nurse cells that support the oocyte by transporting proteins and RNAs. Encapsulated by somatic follicle cells, the cysts develop into egg chambers that progress through 14 stages, culminating in mature, fertilizable eggs [17,18].

To safeguard the fidelity of this process, apoptosis operates at two key checkpoints —within the germarium and during mid-oogenesis —ensuring that only healthy cells contribute to egg development. In the germarium, the first checkpoint triggers apoptotic cell death in response to DNA damage, mediated by p53-induced *hid* expression rather than *rpr* [19,20]. At the second checkpoint during mid-oogenesis, germline cells undergo

caspase-regulated apoptotic and autophagic cell death in nutrient deprived condition, even though RHG genes are dispensable. [21–24]. Although apoptotic cell death is intricately regulated in response to various stimuli during oogenesis, the involvement of *rpr* in germline apoptosis remains largely unknown. Previous studies have identified *stand still* (*stil*) as a critical regulator of *Drosophila* germline cell survival [25]. Genetic interactions with female germline-specific genes, such as *ovo* and *ovarian tumor* (*otu*), along with the female germline-specific requirement of *stil*, suggested its involvement in sex determination [26,27]. However, how Stil controls germline cell fate remains unknown. In this study, we identify *stil* as a repressor of *rpr* in female germline cells. Loss of *stil* function led to apoptosis-mediated depletion of ovarian germline cells, associated with *rpr* upregulation. Stil was enriched at the *rpr* locus and its upstream transcriptional control region, suggesting a direct role in *rpr* regulation. Reporter assays in *Drosophila* S2 cells revealed that Stil represses *rpr* transcription via its N-terminal BED-type zinc finger domain, indicating its role as a repressive transcription factor. In addition, we found that undifferentiated germline stem cell (GSC)-like cells, in which the *rpr* locus exhibits a closed chromatin structure, were resistant to apoptosis caused by *stil* loss. Our findings reveal a molecular mechanism by which Stil and chromatin accessibility repress *rpr*, providing novel insights into the regulation of germline cell fate.

## Results

### Loss of *stand still* (*stil*) leads to female germline cell loss

Previous studies have demonstrated the necessity of *stand still* (*stil*) function in female germline cells; however, its molecular role remains elusive [26,28]. To address this, we analyzed a loss of function allele, *stil^EY16156*, which harbors a P-element insertion near the 3'end of the second exon of *stil* gene, potentially disrupting the splicing of *stil* transcripts (Fig 1A) [29,30]. RT-PCR revealed that *stil* transcripts were undetectable in homozygous *stil^EY16156* ovaries and in *stil^EY16156* testes, indicating a complete loss of *stil* expression in both gonads (Fig 1B).

Similar to the previously reported loss-of-function allele *stil^3*, which exhibited the most severe phenotype (Figs 1A, S1A and S1B) [28], ovaries of both *stil^EY16156* homozygous and *stil^EY16156*/*Df(2R)vg-C* trans-heterozygote, in which the deficiency uncovers the entire *stil* locus, were severely reduced in size and completely lacked germline cells (Fig 1C and 1D). Likewise, *stil^3*/*stil^EY16156* trans-heterozygote females exhibited more severe germline cell loss than *stil^3*/*Df(2R)vg-C* (S1A and S1B Fig). However, male germline cells in *stil^EY16156* testes progressed to differentiate into mature sperms (S1A Fig). Notably, fertility assayed by egg production and hatching rate, was comparable between *stil^EY16156* and *stil^EY16156*/*CyO* males (S1B Fig), indicating that *stil* is specifically required for female germline cells. The loss of germline cells in *stil^EY16156* ovaries was fully rescued by overexpression of the *UASp-6×Myc-Stil* transgene, driven by the germline-specific *NGT40; Nos-Gal4-VP16* driver (Fig 1C and 1D) [31]. Mutant females expressing 6×Myc-Stil, germline cells were differentiated and developed into mature eggs that hatched comparably to those of heterozygous control (Fig 1E). Collectively, these results confirm the pivotal role of *stil* in maintaining female germline cells and ensuring proper oogenesis.

### *stil*-deficient germline cells undergo apoptotic cell death

To investigate the cause of female germline cell loss, we knocked down *stil* function by expressing short hairpin RNA targeting the 4th exon of the *stil* transcript (GL01230) using a germline driver (Fig 1A). Germline-specific knockdown (*stil*-GLKD) reduced *stil* expression in ovaries by approximately 90% compared to control (Fig 2A). *stil*-GLKD females contained significantly smaller ovaries and did not lay any eggs, resembling to *stil* loss-of-function mutant females (Figs 2B and S2A). Unlike the complete germline loss observed in *stil* mutants, however, most ovarioles in *stil*-GLKD retained germline cells, indicating a milder phenotype (Fig 2C). Notably, the number of ovarioles containing at least two egg chambers was remarkably fewer (27.4% versus 100% in the control), suggesting that oogenesis was perturbed before mid-oogenesis (Fig 2B and 2D). While *stil*-GLKD exhibited severe defects similar to its loss-of function alleles, its somatic knockdown using a *tj-Gal*4 driver (*stil*-STKD) did not affect egg production and the hatching rate at all (S2A Fig). These observations suggest that *stil* functions in germline cells both for the survival of female germline cells and egg development.

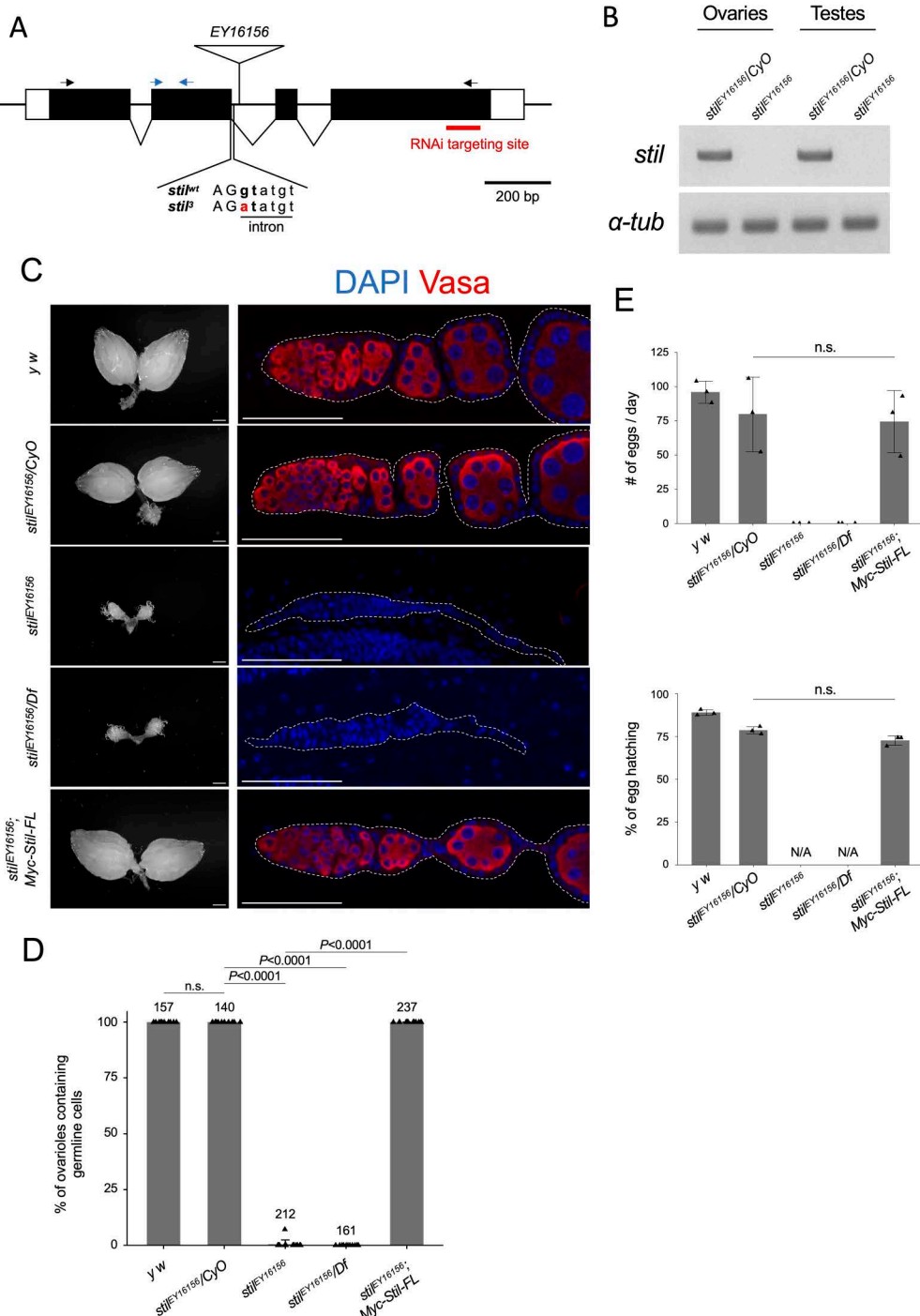

**Fig 1. Disruption of *stil* induces female germline cell depletion. (A)** Schematic representation of the *Drosophila stil* gene locus showing the position of P-element insertion, *EY16156*, mutation site of *stil³*, and the target site of shRNA for knockdown. Black and blue arrows represent primers for RT-PCR and RT-qPCR, respectively. **(B)** Expression of *stil* and the control *a-tub* transcripts in *stil^EY16156* mutant and the heterozygous control ovaries or testes analyzed by RT-PCR. **(C)** Representative images showing entire ovarian morphology (left panels) and ovarioles immunostained with with a germline marker, Vasa (red), and nuclear staining with DAPI (blue) (right panels) from *y w*, *stil^EY16156*/CyO, *stil^EY16156*, *stil^EY16156*/Df, and *stil^EY16156* expressing *6×Myc-Stil-FL* transgene driven by the germline-specific driver, *NGT40; NosGal4-VP16*. Scale bar: 1 mm for whole ovaries in the left panels, 50 μm for ovariole in the right panels. **(D)** Quantification of the percentage of ovarioles containing germline cells per ovary in 2–3-day-old females. Each dot represents

an individual ovary. Genotypes are indicated below the graph and the numbers of ovarioles assessed are shown above each bar. Error bars represent standard deviation (s.d.). **(E)** Analysis of egg laying and hatching rate. The egg numbers and their hatching rate by three females indicated genotypes corresponding to those in **(C)** mated with three *y w* males were measured daily (*n* = 3). Error bars indicate s.d.

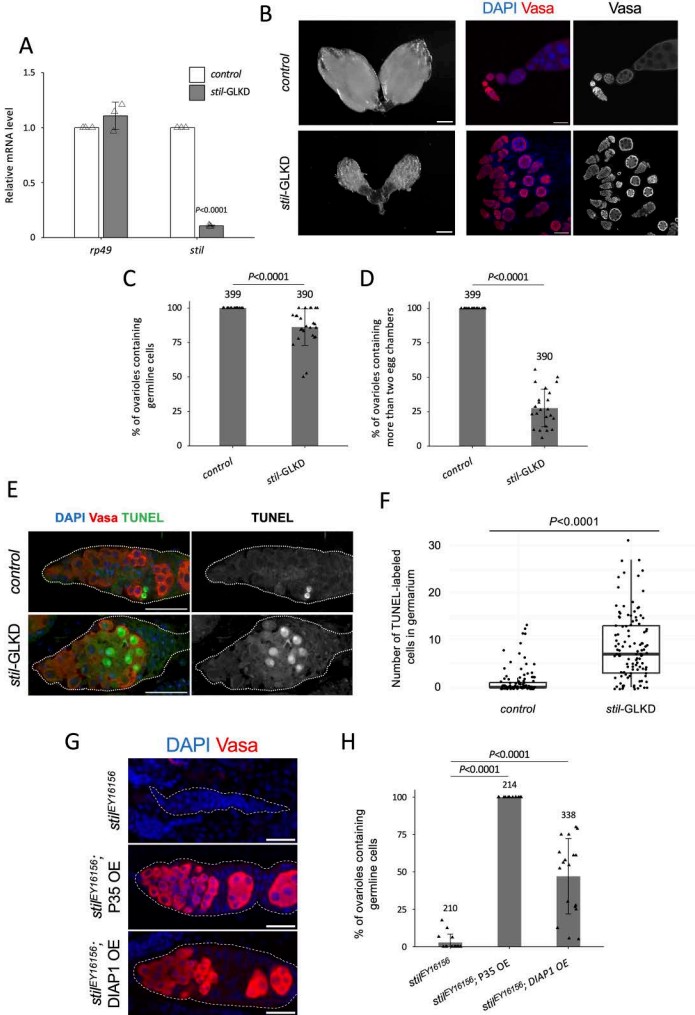

**Fig 2. *stil* loss induces apoptotic cell death in female germline cells. (A)** Quantitative RT-PCR confirming the knockdown efficiency in ovaries of *stil*-germline knockdown (*stil*-GLKD) where shRNA targeting stil transcript was driven by *NGT40; NosGal4-VP16*, and the control with shRNA alone. Expression levels of an internal control, *rp49*, and *stil* are normalized to that of *α-tub*. Error bar indicates s.d. (*n* = 3). **(B)** Representative images showing entire ovarian morphology (left panels) and ovarioles immunostained with a germline marker, Vasa (red), and nuclear staining with DAPI (blue) (right panels) from *stil*-GLKD and the control with shRNA alone. Scale bar: 1 mm for whole ovaries and 50 μm for ovarioles, respectively. **(C)** Quantification of the percentage of ovarioles containing germline cells per ovary in 2–3-day-old females. Each dot represents an individual ovary. Genotypes are indicated below the graph and the number of germarium assessed is noted above each bar. Error bars represent standard deviation (s.d.). **(D)** Quantification of the percentage of developed ovarioles defined as ovarioles that contain at least two egg chambers beyond the germarium per ovary in 2–3-day-old females. Each dot represents an individual ovary. Genotypes are indicated below the graph and the number of germarium assessed is noted above each bar. Error bars represent standard deviation (s.d.). **(E)** Representative images of TUNEL-positive apoptotic cells (green), Vasa (red), and DAPI (blue) in control with shRNA alone and *stil*-GLKD germaria (white dotted). Scale bar: 20 μm. **(F)** Quantification of apoptosis in germarium with TUNEL assay in control and *stil*-GLKD. **(G)** Representative images of germaria (white dotted) from *stil^EY16156^*, *stil^EY16156^; P35 OE (NGT40; NosGal4-VP16 > P35)*, and *stil^EY16156^; DIAP1 OE (NGT40; NosGal4-VP16 > DIAP1)* immunostained with Vasa (red) with DAPI (blue). Scale bar: 20 μm. **(H)** Quantification of the percentage of ovarioles containing germline cells per ovary in 2–3-day-old females. Each dot represents an individual ovary. Genotypes are indicated below the graph and the number of germarium assessed is noted above each bar. Error bars represent standard deviation (s.d.).

Previous studies have suggested that *stil* may play a role in sex determination, particularly through its genetic inter-action with *ovo* and the key sex determination regulator *otu*, which represses the male identity factor Tdrd5l [26,27,32]. To investigate whether *stil* is involved in sex determination, we examined the transcripts of *sxl*, a master regulator of sex determination of germline cells in *stil* mutant ovaries [33]. *sxl* pre-mRNA undergoes alternative splicing to generate sex-specific isoforms, producing a functional Sxl protein essential for female development, or a truncated non-functional male-specific isoform [34,35]. RT-PCR revealed an enrichment of the female-specific *sxl* isoform in both control and *stil*-GLKD ovaries (S2B Fig). In addition, male-specific *sxl* transcripts, detected in both control and *stil*-GLKD testes, were absent in *stil*-GLKD ovaries, indicating that *sxl* splicing patterns remain unaffected by *stil* depletion. These results suggest that *stil* functions independently of sex identity determination in germline cells.

Damaged or abnormal cells are typically removed through regulated cell death mechanisms, including apoptosis, autophagy, and necrosis [36,37]. In *Drosophila* ovaries, such cell death occurs at two distinct developmental checkpoints during oogenesis: in the germarium and during mid-oogenesis, in response to either developmental cues or environmental stressors [38,39]. To determine whether germline cell loss in *stil*-deficient ovaries was due to apoptosis, we performed a TUNEL (Terminal deoxynucleotidyl Transferase Biotin dUTP Nick End Labeling) assay, which detects DNA fragmentation indicative of apoptotic cell death. In region 2A and 2B of the germarium, the first developmental checkpoint in oogenesis [40], the number of TUNEL-positive cells significantly increased in *stil*-GLKD ovaries compared to controls (Fig 2E and 2F). These results suggest that *stil*-GLKD germline cells undergoes apoptotic cell death.

To further confirm the involvement of *stil* in apoptosis, we overexpressed inhibitor of apoptosis proteins (IAPs) in *stil* mutant germline cells. Both the *Drosophila* protein DIAP1 (Death-associated inhibitor of apoptosis 1) and the baculovirus-derived P35 are well-characterized inhibitors that can effectively suppress apoptosis in *Drosophila*. DIAP1 inhibits the activity of initiator caspase, Dronc (Death regulator Nedd2-like caspase) which activates effector caspase, while P35 directly inhibits effector caspase [41–43]. Germline-specific overexpression of P35 fully rescued the germline cell loss observed in *stil*^EY16156 mutants (Fig 2G and 2H). However, germline development remained arrested, and egg chambers degenerated at the mid-oogenesis (S3 Fig). Likewise, DIAP1 overexpression partially restored germline cells (~50% of ovarioles retained germline cells), but did not fully rescue oogenesis defects (Fig 2G and 2H). Altogether, these findings indicate that the loss of *stil* function triggers apoptosis that can be suppressed by apoptosis inhibitors.

## Stil regulates proapoptotic genes to suppress apoptosis in female germline cells

Stil protein harbors a characteristic DNA-binding domain, BED-type zinc finger, containing highly conserved aro-matic residues followed by a shared pattern of cysteines and histidines (S4A Fig) [44]. Given that many of the known BED-type zinc fingers including the founder members, BEAF and DREF, function as transcription factors or chromatin regulators [45–47], Stil may play a role in transcriptional regulation. To examine this possibility, we analyzed the transcriptional profiles of ovarian RNA of *stil* mutant and control. We took advantage of the ectopic expression of P35 to overcome cell death defect of germline cells due to the loss of in *stil* function. The RNA-seq analysis identified 3,569 (30.7%) differentially expressed genes (DEGs) ($\log_2$FC > |1| and p-adj < 0.01) out of 11,608 annotated genes (PRJDB20709; S1 Table). Among these, 2,044 genes were up-regulated (17.6%, red dots) and 1,525 genes were down-regulated (13.1%, blue dots) in *stil* mutant ovaries (S4B Fig). Consistent with the observa-tion of developmental arrest and egg chambers degeneration at the mid-stage of oogenesis (S3 Fig), genes known to be highly expressed in late-stage oogenesis, such as proteins composing the chorion, were significantly down-regulated in *stil* mutant ovaries (Fig 3A). Of those upregulated, notably, the pro-apoptotic gene *rpr* (*reaper*) ranked among the top 10 upregulated DEGs (Fig 3A). Given the apoptotic phenotype of *stil* mutant germline cells, 67 apoptosis-related genes out of 224 annotated in the dataset (GO:0006915) were identified as DEGs (Fig 3B). Among the top 10 genes list of apoptosis-related DEGs, pro-apoptotic genes, *rpr* and *hid* (*head involution defec-tive*), were significantly upregulated, while the hedgehog signaling genes expressed in later oogenesis such as

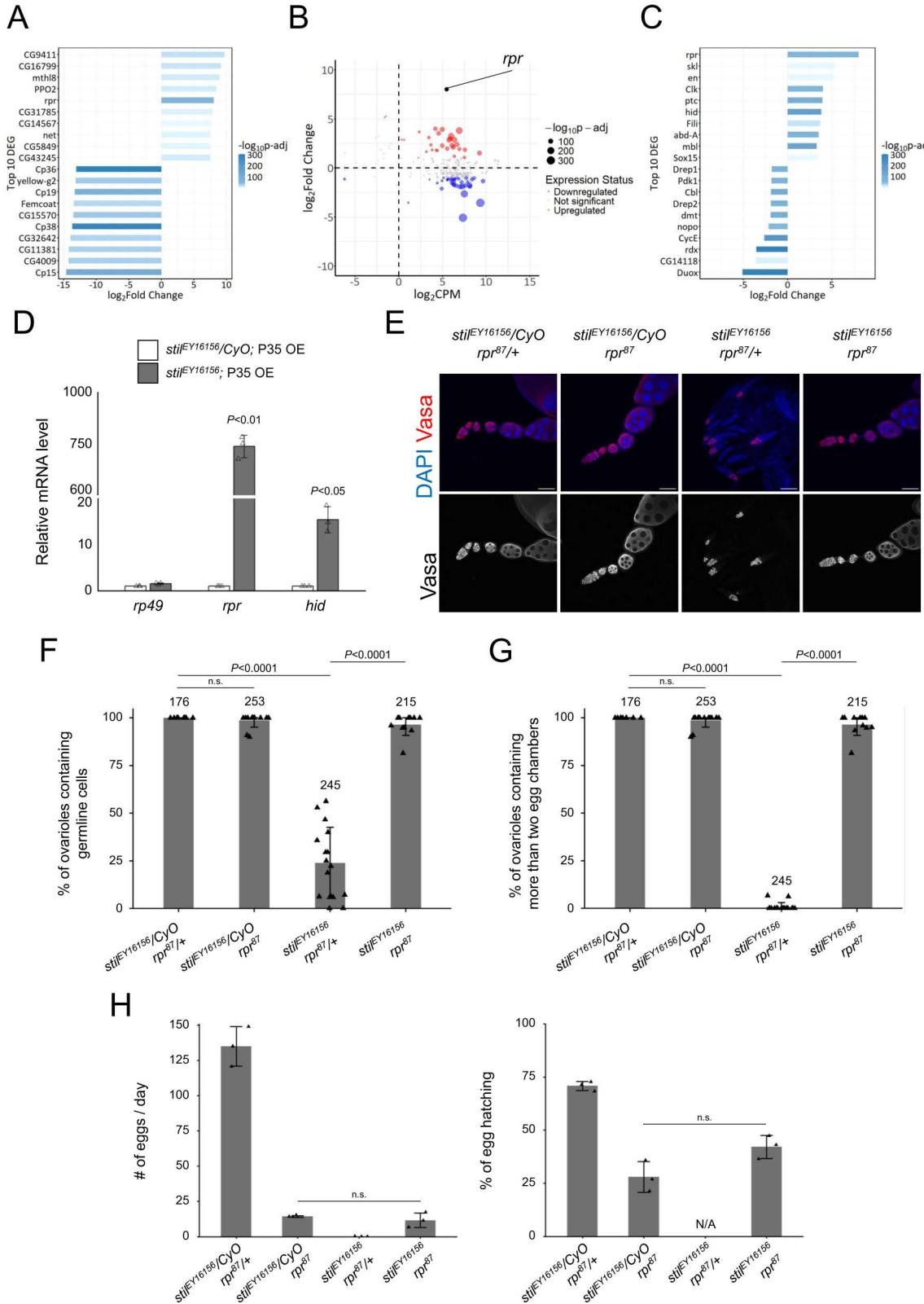

**Fig 3. *stil* regulates *rpr* expression in female germline cells. (A)** Top 10 upregulated and downregulated genes in *stil*-deficient ovaries. Bars indicate fold changes in *stil* mutant ovaries overexpressing (OE) the apoptosis inhibitor, P35 in germline cells (*stil^EY16156^, NGT40; NosGal4-VP16>P35*), compared

to control with P35 OE in a heterozygous background (*stil^{EY16156}/CyO, NGT40; NosGal4-VP16 > P35*). Bars are colored according to adjusted p-values (p-adj). **(B)** MA plot of apoptosis-related genes (GO:0006915) showing their mean gene expression (CPM: counts per million) and log2 fold change in *stil^{EY16156}; P35 OE* comparing *stil^{EY16156}/CyO; P35 OE*. Each dot represents an individual gene, with dot size inversely correlated with p-value adjusted (p-adj). Larger dots highlight genes with higher statistical significance. The red dots (upregulated genes) and blue dots (downregulated genes) represent the differentially expressed RNAs with fold threshold>2.0, and p-adj < 0.01. **(C)** Top 10 upregulated and downregulated genes related to apoptosis (GO:0006915). Bars represent fold changes and are colored by adjusted p-values (p-adj), as in (A). **(D)** Quantitative RT-PCR analysis of pro-apoptotic genes, *rpr* and *hid* in *stil^{EY16156}; P35 OE* and *stil^{EY16156}/CyO; P35 OE* ovaries. Expression levels were normalized to *a-tub* with *rp49* as an internal control. Error bars indicate s.d. (n = 3). **(E)** Immunofluorescence of Vasa (red) and DAPI (blue) in ovarioles from *stil^{EY16156}/CyO; rpr^{87}/+, stil^{EY16156}/CyO; rpr^{87}, stil^{EY16156}; rpr^{87}/+, and stil^{EY16156}; rpr^{87}*. Scale bar: 50 μm. **(F)** Quantification of the percentage of ovarioles containing germline cells per ovary in 2–3-day-old females. Each dot represents an individual ovary. The number of germarium assessed are noted above each bar. Error bars represent standard deviation (s.d.). **(G)** Quantification of the percentage of developed ovarioles defined as ovarioles that contain at least two egg chambers beyond the germarium per ovary in 2–3-day-old females. Each dot represents an individual ovary. The number of germarium assessed are noted above each bar. Error bars represent standard deviation (s.d.). **(H)** Analysis of egg laying and hatching rate. Daily egg laying by three females indicated genotypes mated with three *y w* males and their hatching rates are shown (n = 3). Error bar indicates s.d.

*duox* and *rdx* were significantly downregulated (Fig 3C). *rpr* and *hid* are the key members of IAP antagonists, known for their central role in initiating apoptosis [1,4]. Their activation through various signaling pathways leads to the ubiquitin-mediated degradation and translational inhibition of DIAP1, which subsequently results in the activation of Caspase proteins [6,48]. Further validation by RT-qPCR demonstrated that *rpr* expression increased approximately 700-fold, while *hid* expression increased 15-fold in the rescued *stil* mutant ovaries by P35 expression (Fig 3D). To exclude a possibility that egg chamber degeneration in germline-rescued *stil* mutants affects pro-apoptotic gene expression, we isolated RNAs from early-stage ovaries (up to stage 5–6 of oogenesis) by manual removal of later stages and re-examined their expression levels. *rpr* expression remained elevated in RNAs prepared from early-stage ovaries, while *hid* expression did not (S5A Fig). Furthermore, RT-qPCR analysis using *stil*-GLKD ovaries driven by three different germline drivers, *NGT40; NosGal4-VP16*, *Bam-Gal4*, and *mata-Tub-Gal4*, consistently exhibited significant upregulation of *rpr* but not *hid* across all conditions (S5B Fig). Conversely, *rpr* was not prominently upregulated in *stil*-GLKD testes with *NGT40; NosGal4-VP16* or *Bam-Gal4*, consistent with the phenotypic observations in *stil* mutant testes (S5C Fig). These results suggest that *rpr* is robustly upregulated in the absence of *stil* function in female germline cells.

To determine whether pro-apoptotic genes upregulation directly contributes to apoptosis of *stil* mutant female germline cells, we generated double mutants of *stil* and the proapoptotic gene, *rpr*. Strikingly, the germline cell death and differentiation defects observed in *stil* mutant ovaries were fully rescued in the double mutant ovaries (Fig 3E–G). Germline development proceeded, resulting in the formation of mature eggs and partial restoration of fertility; mutant females laid a few eggs and approximately 40% of those hatched (Fig 3H). In contrast, *stil-hid* double mutants showed a slight increase in germline cell survival, but no developing egg chambers were observed (S5D and S5E Fig), indicating that *hid* does not play a central role in apoptosis caused by the loss of *stil* function. The complete rescue of germline survival in *stil rpr* double mutants also suggests that the failure of P35 overexpression to restore mid-oogenesis defects may partly reflect insufficient transgene expression (S3 Fig). Alternatively, Rpr accumulation in *stil* mutants might activate an *rpr*-driven cascade that results in egg chamber degeneration. Together, these findings highlight that *rpr* regulation by *stil* is critical for suppressing apoptosis in female germline cells, establishing *rpr* as a key downstream effector in this pathway.

## Stil represses *rpr* expression by targeting the regulatory regions via its BED-type zinc finger

The transcription of *rpr* is governed by a complex network of developmental, environmental, and stress-related signals, each employing distinct regulatory mechanisms. These signals modulate transcription factors such as p53, Dfd, the EcR/USP complex, AP-1, Shn, and abd-A leading to either the activation or suppression of apoptosis [8–11,49]. These transcription factors interact with upstream enhancer and promoter regions of the *rpr* locus (*rpr*-transcriptional control region) to either activate or suppress its transcription, ensuring precise spatial and temporal control of apoptosis [36]. To examine

whether Stil also directly interacts with *rpr*-transcriptional control region, we performed DNA adenine methyltransferase identification (DamID)-seq analysis [50]. A fusion of Stil with *E. coli* DNA adenine methyltransferase (Dam-Stil) or Dam alone as a control were expressed specifically in germline cells [51,52]. Genomic DNA from ovaries was digested with DpnI and subjected to NGS to detect GATC sites methylated by Dam-Stil near Stil-bound regions (PRJDB20722). Genome-wide DamID-seq analysis revealed that ~80% of Stil-enriched peaks were located in promoter regions, supporting a role for Stil in transcriptional regulation (Fig 4A). A total of 682 genes with Stil-enriched peaks detected at promoter regions (≤1 kb) showed significantly altered expression in RNA-seq analysis of *stil* mutants expressing P35, including *rpr* (S4 Table). At higher resolution, prominent peaks were observed at the *rpr* locus and its upstream region (5–6 kb), suggesting direct binding of Stil to the *rpr* regulatory element (Fig 4B). Next, we examined the role of Stil's BED-type zinc finger in *rpr* repression by generating four Stil variants: Stil-FL (full-length), Stil-NT (N-terminal domain containing the BED-type zinc finger and nuclear localization signal), Stil-CT (C-terminal domain lacking these two motifs), and Stil-AAYA (full-length Stil with a mutated BED-type zinc finger substituting four residues including two critical cysteines) (Fig 4C). The expression of all four Stil variant proteins from the transgenes was confirmed, although Stil-CT showed a slightly reduced expression level (S6A Fig). In nurse cells, despite lacking the predicted nuclear localization signal, Stil-CT localized to the nucleus and overlapped with chromosomes, as did Stil-FL and Stil-AAYA, whereas Stil-NT showed diffuse nuclear distribution without being enriched onto chromosomes (S6B Fig). Collectively, these data suggest that Stil's nuclear and chromatin localization are mediated by the C-terminal domain, independently of the zinc finger motif. To assess the functional significance of these domains *in vivo*, we expressed the variants in *stil^EY16156* germline cells. Stil-FL fully rescued the defects of both germline cell survival and development, leading to the production of mature eggs (Fig 4D-4F). Remarkably, Stil-NT partially rescued germline defects despite its diffuse nuclear localization, while neither Stil-CT nor Stil-AAYA, both lacking functional BED-type zinc fingers, restored germline cell survival or differentiation. Although CT shows slightly lower expression (S6A Fig), AAYA fails to rescue despite FL-like expression, indicating that expression level is not limiting and that loss of the BED-type zinc finger underlies the phenotype. The relatively stronger rescue observed with AAYA compared to CT may be attributable to its higher expression level or partial contributions from the N-terminal region lacking the BED-type zinc finger (Fig 4D–4F). These findings suggest that the BED-type zinc finger is essential for Stil function in germline cells possibly as a transcription regulator, while the C-terminal domain mediates nuclear and chromatin localization.

To validate whether the BED-type zinc finger functions in *rpr* repression through interaction with the *rpr* transcriptional control region, we performed a reporter assay using *Drosophila* S2 cells. We constructed a *rpr*-GFP reporter containing the 5-kb upstream region of *rpr*, where DamID-seq signals were detected, fused to GFP (Fig 4G) and transfected with Stil variants. Stil variants exhibited localization patterns in S2 cells similar to those in nurse cells (S6C Fig), supporting their suitability for functional analysis in S2 cells. Consistent with the *in vivo* rescue experiment (Fig 4D–4F), co-expression of Stil-FL with the *rpr*-GFP reporter significantly repressed GFP expression. Stil-NT containing the BED-type zinc finger also reduced GFP expression, albeit partially. In contrast, Stil-CT and Stil-AAYA failed to exhibit any detectable repressive effects (Fig 4H). To confirm the specificity of this Stil-mediated repression, we utilized a *ubiP*-GFP reporter containing the *ubi-p63E* gene promoter fused to GFP (S6D Fig). Co-expression of Stil-FL with the *ubiP*-GFP reporter showed no reduction in GFP expression (S6E Fig), suggesting that Stil preferentially targets the *rpr* upstream region. These results demonstrate that Stil interacts with the *rpr* transcriptional control region via its BED-type zinc finger, which is critical for transcriptional repression, while the chromatin association through the C-terminal region is required for full repressive activity in germline cells.

## Undifferentiated germline cells exhibit resistance to apoptosis in *stil* loss

Loss of *stil* function resulted in severe depletion of germline cells by the 3rd instar larval stage (120 hours after egg laying, AEL), leaving only a few surviving germline cells in *stil* mutant (Fig 5A and 5B). Notably, these surviving germline cells were predominantly found near the terminal filaments and future cap cells, the niche that supports germline stem cell

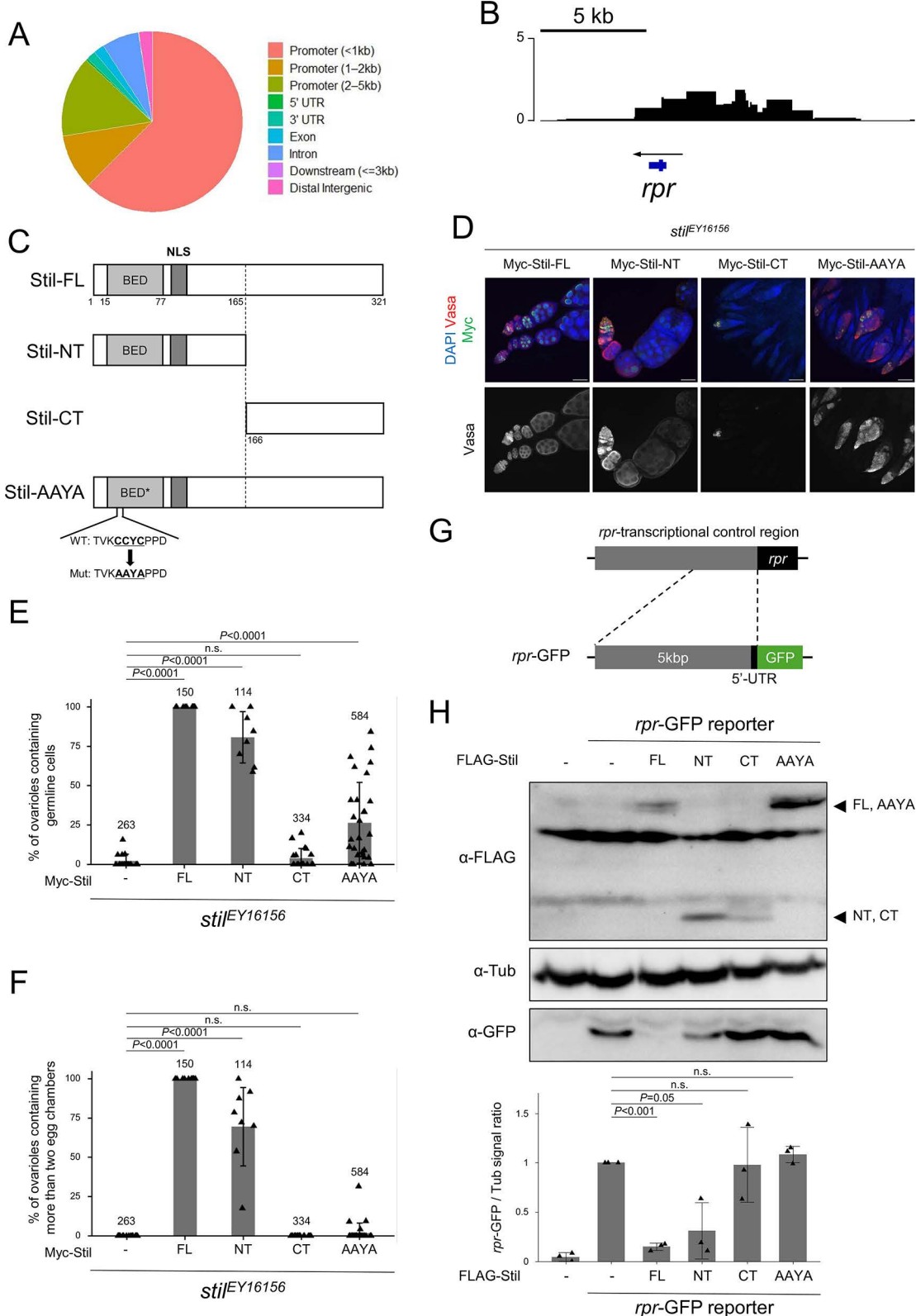

**Fig 4. Stil interacts with *rpr*-transcriptional control region to repress its expression. (A)** Distribution of DamID peaks across entire genome categorized by genomic features including promoter (≤1, 1–2, and 2–3 kb), UTR, exon, intron, distal intergenic, and downstream. **(B)** The plot showing the

log$_2$ ratio of Dam-Stil peaks to Dam-alone control across the *rpr* genomic region. The blue bar on the bottom indicates the *rpr* gene body, and the arrow shows the direction of *rpr* transcription. Three independent biological replicates were performed for both Dam-Stil and Dam-alone and data were merged for visualization. **(C)** Schematic representation of the Stil full-length (Stil-FL), the truncated variants: Stil-NT (1$^{st}$ to 165$^{th}$ aa) and Stil-CT (166$^{th}$ to the end), and amino acid substitutions of the C2H2 zinc finger CCYC: Stil-AAYA. **(D)** *stil*$^{EY16156}$ ovaries expressing Myc-tagged Stil variants corresponding to (C) in germline cells. Ovaries are immunostained with a germline marker, Vasa (red), and nuclear staining with DAPI (blue). Scale bar: 50 µm. **(E)** Quantification of the percentage of ovarioles containing germline cells per ovary in 2–3-day-old females. Each dot represents an individual ovary. Genotypes are indicated below the graph and the number of germarium assessed is noted above each bar. Error bars represent standard deviation (s.d.). **(F)** Quantification of the percentage of developed ovarioles defined as ovarioles that contain at least two egg chambers beyond the germarium per ovary in 2–3-day-old females. Each dot represents an individual ovary. Genotypes are indicated below the graph and the number of germarium assessed is noted above each bar. Error bars indicate standard deviation (s.d.). **(G)** Schematic representation of *rpr*-GFP reporter containing the 5 kb upstream region and 5'-UTR of the *rpr* gene fused to GFP CDS. **(H)** Reporter assay in S2 cells followed by western blot analysis to detect GFP and α-Tubulin (αTub) as a loading control. Co-transfection of the *rpr*-GFP reporter and FLAG-tagged Stil variants were performed in S2 cells. Arrowheads indicate Stil variant proteins. The lower panel shows the signal ratio of GFP expression to αTub, normalized to that in cells transfected with the reporter alone. Error bars represent the standard deviation (s.d.) (*n* = 3).

(GSC) (Fig 5A) [18]. This observation suggested that GSCs within the niche may be resistant to cell death caused by *stil* loss. To test whether GSCs are inherently resistant to cell death caused by *stil* deficiency, we induced GSC-like properties by overexpressing a constitutively active the Decapentaplegic (Dpp) receptor, Thickveins (Tkv.CA) in germline cells, or the loss of function of *bag of marbles* (*bam*), a key differentiation factor [53,54]. These manipulations resulted in the tumorigenic proliferation of undifferentiated GSC-like cells or pre-cystoblast, respectively [55]. Remarkably, Tkv.CA overexpression and *bam* loss rescued germline cell depletion in *stil* mutant ovaries, leading to the formation of tumorous germaria containing undifferentiated germline cells comparable to the *stil* heterozygous control (Fig 5C and 5D). Indeed, TUNEL assay showed no increase in apoptotic cells in such tumorous germaria (Fig 5E and 5F), indicating that the undifferentiated state of germline cells confers resistance to apoptosis in the absence of *stil*.

Consistent with this observation, RNA-seq and gene Ontology (GO) analysis of those tumorous ovaries revealed that the expression levels of most apoptosis-related genes (GO:0006915), including *rpr*, remain unaffected (PRJDB20709; S2 Table) (Fig 5G). Furthermore, RT-qPCR confirmed that *rpr* expression was not significantly upregulated in the GSC-like tumor *stil* mutant ovaries compared to P35-rescued *stil* mutant ones (Figs 3D and 5H). Compared to GSCs, which were almost completely lost in *stil* mutants, GSC-like cells may retain a more robust stemness owing to the extremely elevated Dpp signaling pathway, potentially resulting in stronger repression of *rpr* expression. These results suggest that apoptosis-related genes, including *rpr*, are transcriptionally repressed in GSC-like cells even in the absence of *stil* function.

### *rpr* silencing in undifferentiated germline cells is associated with chromatin states

Previous studies in mammals have established that stem cells are generally resistant to apoptosis. In mouse mammary stem cells and cancer stem cells, this resistance is attributed to several mechanisms, including enhanced DNA repair and overexpression of anti-apoptotic proteins [56–58]. Similarly, in *Drosophila*, intestinal stem cells were shown to resist cell death through a two-tiered regulation of *rpr*, involving chromatin state and spatial control of its expression [59]. To explore whether chromatin dynamics contribute to GSC apoptosis resistance in *stil* mutants, we analyzed the chromatin status at the *rpr* locus using Assay for Transposase-Accessible Chromatin-seq (ATAC-seq) data of *Drosophila* ovarian cells [60]. This dataset was generated using fluorescence-activated cell sorting (FACS) of GSCs (2C–4C) and nurse cells (NCs) at various stages of egg chamber maturation, with ploidy levels ranging from 4C to 512C due to endoreplication [61]. In early mid-stage NCs where *stil* deficient-germline cells undergo apoptosis, ATAC-seq signals were enriched at three upstream regions of *rpr* (-0.5 to 0.5 kb, 1.5 to 2.0 kb, and 3.5 to 5.0 kb relative to the transcription start site [TSS] of *rpr*) (Fig 6A). Conversely, signals at these three regions were significantly lower in 2C-4C GSCs and 64C-512C NCs. These results suggest that the chromatin state upstream of *rpr* is closed in GSCs and late-stage NCs (64C-512C) but open in early mid-stage NCs (4C-32C). Notably, despite

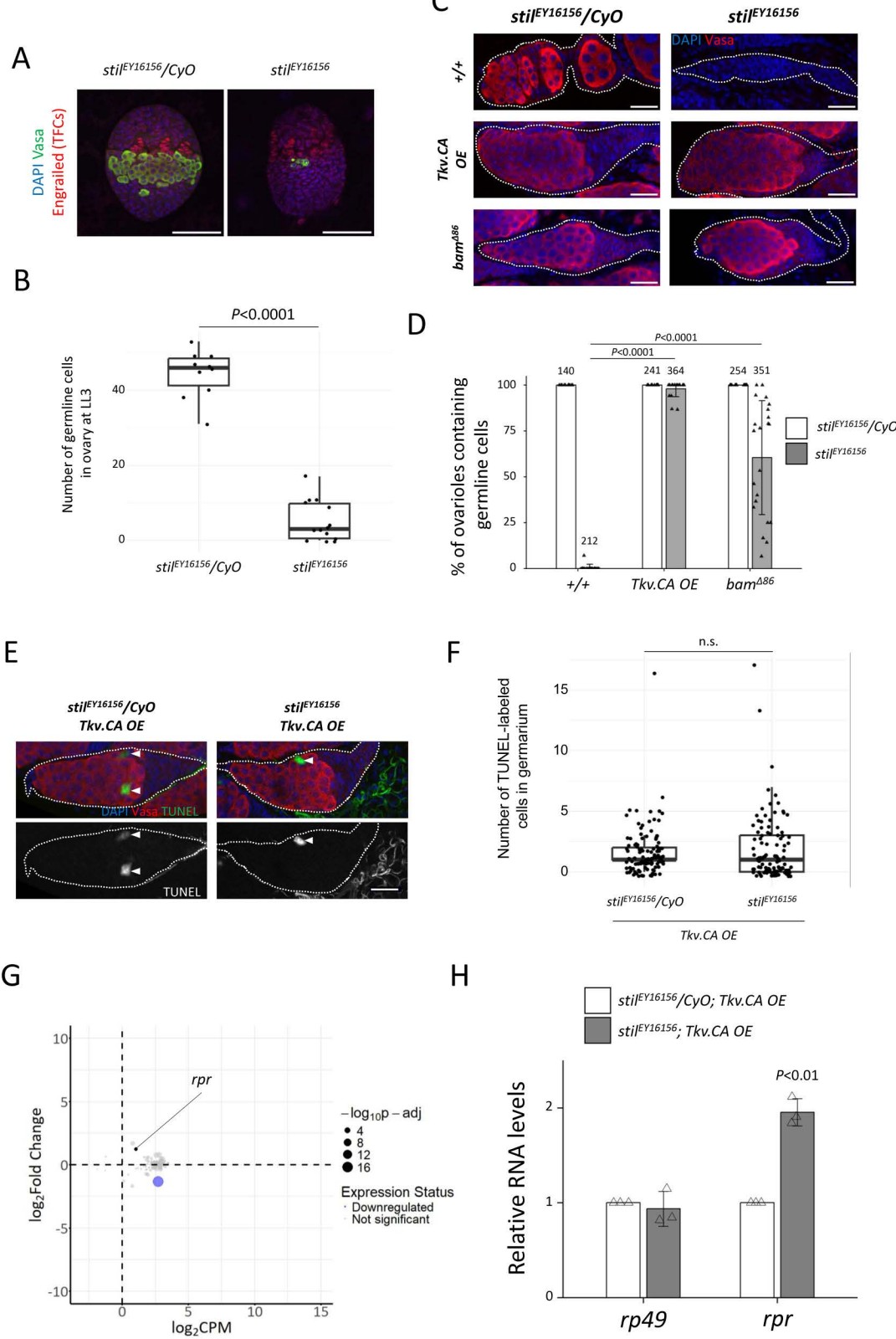

**Fig 5. *stil* mutant germline stem cells resist apoptosis. (A)** Fluorescent immunostaining with anti-Engrailed (red), anti-Vasa (green), and DAPI (blue) for terminal filaments of late 3rd instar larval (LL3) ovaries from the heterozygous control, *stil^EY16156^/CyO* and *stil* mutant, *stil^EY16156^*. Scale bar: 50 μm. **(B)**

PLOS Genetics

Quantification of surviving germline cells in 3rd instar larval ovaries from *stil*[EY16156]/*CyO* and *stil*[EY16156]. **(C)** Immunostaining of Vasa (red) and DAPI (blue) in control, *Tkv.CA* (constitutive active Thickveins) *OE* (*NGT40; NosGal4-VP16 > Tkv.CA*), and *bam* mutant (*bam*[Δ86]) germaria (white dotted) in the control heterozygous (*stil*[EY16156]/*CyO*) or *stil* mutant (*stil*[EY16156]) background. Scale bar: 20 μm. **(D)** Quantification germline cell-containing germaria per ovary in 2–3 days-old females. Each dot represents an individual ovary. Genotypes and the number of germarium assessed are indicated above each bar. Error bars represent standard deviation (s.d.). **(E)** Representative images of germaria (white dashed lines) showing TUNEL-positive apoptotic cells (green), Vasa (red), and DAPI (blue) in the control tumorous germaria (*stil*[EY16156]/*CyO; Tkv.CA OE*) and tumorous germaria of *stil* mutant ovaries (*stil*[EY16156]; *Tkv. CA OE*). White arrowheads indicate TUNEL-positive apoptotic cells. Scale bar: 20 μm. **(F)** Quantification of apoptosis in germaria from the control tumorous ovaries (*stil*[EY16156]/*CyO; Tkv.CA OE*) and *stil* mutant ovaries (*stil*[EY16156]; *Tkv.CA OE*). **(G)** MA plot of apoptosis-related genes showing the mean gene expression (CPM: counts per million) and log2 fold change in *stil* mutant ovaries (*stil*[EY16156]; *Tkv.CA OE*) compared to the control tumorous ovaries (*stil*[EY16156]/*CyO; Tkv.CA OE*). Each dot represents an individual gene, with dot size inversely correlated with p-adj. The blue dots represent the downregulated genes with fold change < -2.0, and p-adj < 0.01. **(H)** Quantitative RT-PCR analysis of *rpr* in the control tumorous ovaries (*stil*[EY16156]/*CyO; Tkv.CA OE*) and *stil* mutant ovaries (*stil*[EY16156]; *Tkv.CA OE*). Expression levels of an internal control, *rp49*, and *rpr* are normalized to that of *α-tub*. Error bars indicate s.d. (*n* = 3).

harboring the same ploidy, 4C GSCs and 4C NCs displayed distinct chromatin accessibility, reflecting regulatory differences likely responsible for their differential responses to apoptosis (Fig 6A). We further analyzed epigenetic marks such as H3K9me3 and H3K27me3 at the *rpr* locus using publicly available data sets [60,62]. As internal controls, we confirmed H3K9me3 enrichment at the *light* (*lt*) locus and H3K27me3 enrichment at the *Ultrabithorax* (*Ubx*) locus, consistent with their established chromatin states [63–65]; relative to these controls, the *rpr* locus shows H3K9me3 but no clear H3K27me3 enrichment in GSCs. H3K9me3, a histone modification associated with transcriptional repression, was observed around the *rpr* locus and its transcriptional control region in GSCs, whereas only minimal signals were detected in 4C NCs (Fig 6B). In contrast, H3K27me3, another repression mark exhibited no difference between GSCs and 32C NCs (Fig 6C) [62]. This suggests that H3K9me3, rather than H3K27me3, may establish a heterochromatic state at the *rpr* loci in GSCs, contributing to transcriptional silencing. These findings imply that chromatin dynamics play a crucial role in regulating *rpr* expression during germline development.

## Discussion

In this study, we investigated the role of the *Drosophila* gene *stand still* (*stil*), which is required for the survival and maturation of female germline cells. Loss of *stil* leads to apoptosis of germline cells via upregulation of the pro-apoptotic gene *rpr* (Fig 3E), identifying *rpr* as the primary effector of cell death in this context. GFP-reporter assays with Stil variants in S2 cells suggest that Stil represses *rpr* expression, likely through direct binding to its upstream regulatory region via the BED-type zinc finger domain (Fig 4H).

The transcription of *rpr* is regulated by multiple transcription factors that bind to cis-regulatory elements located upstream of the gene [36]. For example, the tumor suppressor protein p53 binds to a well-characterized p53 response element (p53RE) located 4.8 kb upstream of *rpr*, which is responsive to intrinsic and extrinsic signals during embryogenesis and in germline cells [66,67]. However, p53 binding to this element has only been observed in the anterior germarium, and no direct link between p53 and apoptosis has been established in the female germline [20]. Other known regulators, such as the homeobox transcription factor Deformed (Dfd), abdominal A (abd-A) and the JNK-responsive complex AP-1, also bind upstream *rpr* enhancers in somatic tissues, but neither is active in the female germline [8,11,49]. Thus, factors that suppress *rpr* transcription specifically in germline cells have remained unidentified. Our study identifies Stil as a female germline-specific repressor of *rpr*. DamID profiling revealed Stil binding to the *rpr* locus to 5–6 kb upstream, covering regulatory elements associated with known transcription factors (Fig 4B). Consistently, Stil repressed *rpr*-GFP reporter containing the 5 kb upstream regulatory region in S2 cells, and this repression required Stil's BED-type zinc finger domain (Fig 4H). This domain is known to mediate DNA binding and contribute to transcriptional regulation and chromatin remodeling [44]. These findings support a model in which Stil directly occupies the *rpr* regulatory region and blocks activator access to maintain transcriptional repression in germline cells. Notably, the DamID peak intensity at the *rpr* locus reached 3.41,

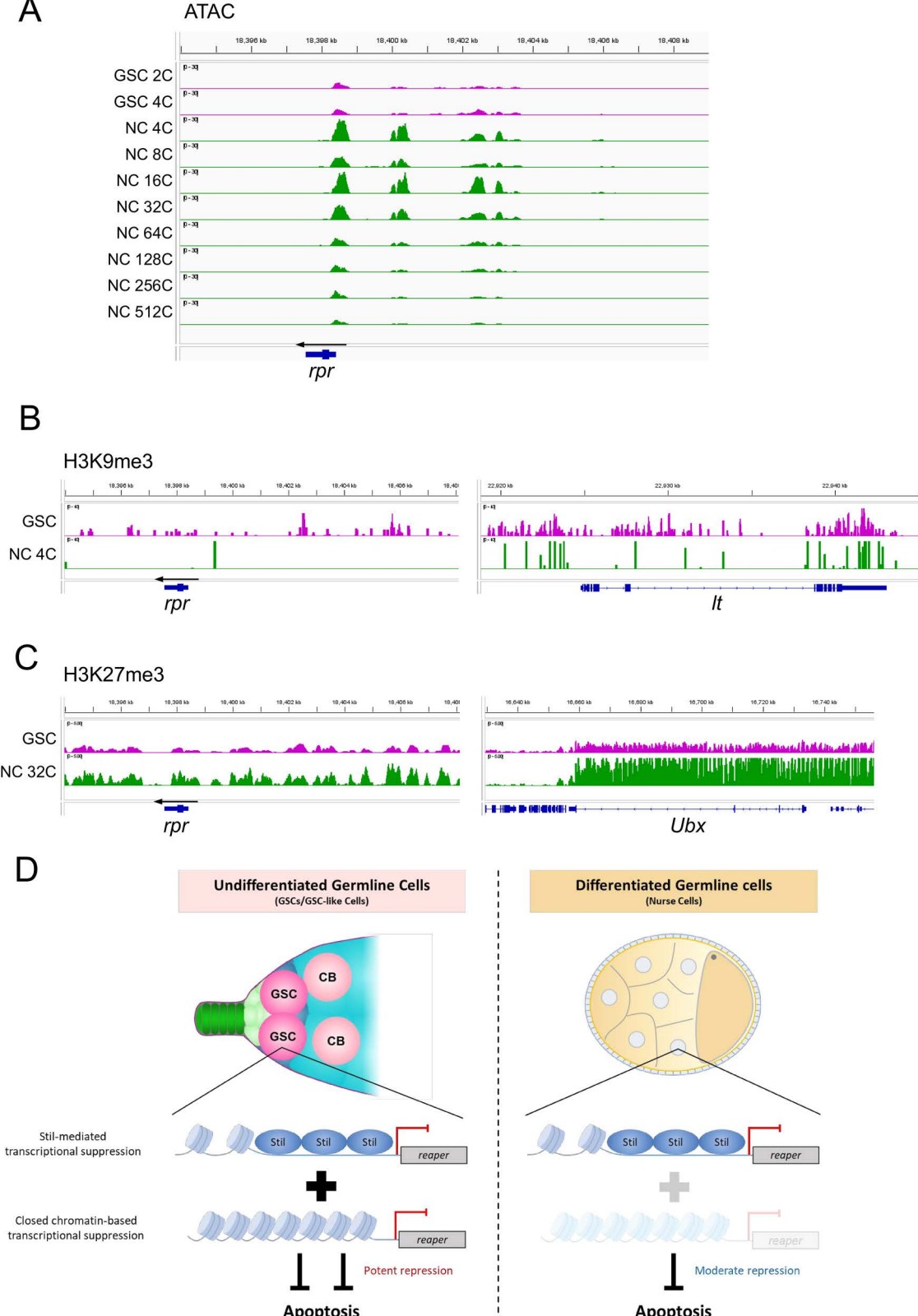

**Fig 6. Chromatin state changes in female germline cells during oogenesis. (A)** ATAC-seq signal intensity (RPM-normalized) at the *rpr* locus in germline stem cells (GSCs: 2N and 4N) and nurse cells (NCs) at different ploidy levels (4C to 512C). The Y-axis is fixed at 0–30 RPM for all tracks. For

64C, 128C, 256C, and 512C NCs, data from three biological replicates are averaged; all other samples are based on single replicates. **(B)** H3K9me3 ChIP-seq signal at the *rpr* locus and the *lt* locus (H3K9me3-positive control) in GSCs and 4C NCs. The Y-axis is fixed at 0–40 RPM for all tracks. **(C)** H3K27me3 ChIP-seq signal at the *rpr* locus and the *Ubx* locus (H3K27me3-positive control) in GSCs and 32C NCs. The Y-axis is fixed at 0–5 RPM for all tracks. **(D)** Schematic model illustrating the *rpr* repression to inhibit the apoptotic cell death in undifferentiated germline cells (GSCs or GSC-like cells) and differentiated germline cells (nurse cells).

which is moderate rather than strong (S4 Table). This suggests that, in addition to repressing *rpr*, Stil may bind to and regulate other genomic loci in the female germline. Investigating the repertoire of Stil target genes and elucidating their roles in germline cells will be an important future direction of this study.

Whereas DIAP1 overexpression only partially rescued germline loss in *stil* mutants, P35 restored germline survival in nearly all cases (Fig 2G and 2H), highlighting a functional distinction in how these inhibitors act in the context of *rpr*-induced apoptosis. Both inhibit apoptosis through distinct mechanisms: DIAP1 blocks caspase activation by inhibiting initiator and effector caspases, while P35, a baculoviral protein, acts as a pseudo-substrate that irreversibly inhibits effector caspases such as DrICE and Dcp-1 [68,69]. *rpr* antagonizes DIAP1 through two independent mechanisms—by promoting its auto-ubiquitination and proteasomal degradation, and by inhibiting its translation [6,48]. Therefore, high *rpr* levels in *stil* mutant ovaries may deplete functional DIAP1, limiting the rescue effect of its overexpression. By contrast, P35 directly targets downstream caspases, bypassing DIAP1 loss and producing stronger inhibition. These results support the model that Stil maintains the apoptotic threshold by regulating *rpr* levels and balancing Rpr–DIAP1 interactions [7,70].

Despite the loss of *stil*, undifferentiated female germline cells (GSCs and pre-cyst cells) remain resistant to apoptosis (Fig 5E and 5F), correlating with closed chromatin and H3K9me3 enrichment at the *rpr* locus (Fig 6A and 6B). Similar chromatin-based repression of *rpr* has been reported in *Drosophila* intestinal stem cells [59] and in late-stage embryos, where heterochromatin silences *rpr* expression [12]. Together with our findings, these studies suggest that chromatin-mediated silencing provides an additional layer of protection against apoptosis in female GSCs, independent of *stil*. By contrast, *stil* mutant male germline cells show no phenotype and no *rpr* upregulation (S5B Fig), suggesting that that *rpr*-mediated apoptosis is not activated in the male germline. This points to a sex-specific mechanism of *rpr* repression, potentially involving chromatin-based silencing, although this possibility has yet to be examined.

*rpr* is conserved only within Diptera, although its IAP-binding motif, essential for apoptosis induction, is broadly conserved across metazoans [71–78] (S7 Fig). Similarly, *stil* is also restricted to Diptera, predominantly within *Drosophila*, whereas its BED-type zinc finger domain is widely conserved among diverse organisms [44,79–81]. Phylogenetic patterns across Diptera are consistent with a model in which *stil* acts as a dedicated repressor of *rpr* in the *Drosophila* germline cells (S7 Fig). Due to its potent pro-apoptotic activity, *rpr* must be stringently repressed in a spatiotemporal manner through mechanisms that are specific to both cell type and developmental stage. During embryogenesis, repression of *rpr* is mediated by the Dpp-signaling factor Shn, which binds to the *rpr* regulatory region, whereas in intestinal stem cells (ISCs), its expression is suppressed through chromatin conformation. In *Drosophila* female germline cells, *hid* serves as the primary regulator of apoptosis, while *rpr* activity is generally suppressed [20,24]. However, *rpr* mutants exhibit reduced fertility despite producing viable eggs (Fig 3H), suggesting that *rpr*-mediated apoptosis may be required for proper egg development. Accordingly, we propose that *stil* restrains *rpr* in the *Drosophila* female germline, allowing *hid* to predominate in apoptotic regulation.

Together, our findings reveal two mechanisms that protect female germline cells from *rpr*-dependent apoptosis: Stil-mediated transcriptional repression, which operates in both differentiated and undifferentiated germline cells, and chromatin-based silencing, which functions specifically in undifferentiated germline cells (Fig 6D). Although the upstream regulation of *stil* is unknown, its selective role in female germline cells suggests it functions as a developmental checkpoint

that prevents premature apoptosis during oogenesis. A key question is how *stil* integrates intrinsic or environmental cues to modulate the apoptotic threshold during germline development.

## Materials and methods

### Fly stock and cultures

All *Drosophila* stocks and crosses were reared at room temperature (25ºC) on standard food [5% (w/v) dry yeast, 5% (w/v) corn flower, 2% (w/v) rice bran, 10% (w/v) glucose, 0.7% (w/v) agar, 0.2% (v/v) propionic acid and 0.05% (w/v) p-hydroxy butyl benzoic acid]. The following stocks were used in this study: *yellow white* (*y w*), *attP2* (BL#25710), *stil^EY16156* (BL#21191), *stil^3* (BL#533, splicing donor site of the second intron was mutated from GTATG to ATATG at 12570993), *Df(2R)vg-C* (BL#754, deficiency for *stil*), *Df(3L)X14* (BL#600368, deficiency for *hid*) *UASp-P35* (a gift from Dr. Bergmann), *UASp-DIAP1* (BL#63818), *rpr^87* (BL#83150), *hid^05014* (BL#83349), *UASp-Tkv.CA* (a gift from Dr. Dahua Chen), *bam^Δ86* (BL#5427), *NGT40-Gal4*; *nosGal4-VP16* [31], *mat α-tub-Gal4* (BL#7063), *bam-Gal4* [82], and *tj-Gal4* (DGRC#104055). The mutation site of *stil^3* was identified by sanger sequencing of the PCR-amplified fragment.

### Generation of transgenic fly lines

Transgenic fly lines expressing Stil variants (6×Myc-Stil-FL, 6×Myc-Stil-NT, 6×Myc-Stil-CT, and 6×Myc-Stil-AAYA) and DamID constructs (Dam-alone and Dam-Stil) were generated using the PhiC31 integrase-mediated transgenesis system. DNA fragments encoding Stil variant were amplified using cDNAs obtained by reverse transcription of ovarian RNA extracted from *y w* flies. Six tandem Myc tags were appended to the N-terminal region of the amplified Stil variant fragments through ligation. DamID constructs were designed such that a primary open reading frame (ORF) encoding GFP was followed by two stop codons and a secondary ORF encoding either Dam or Dam-Stil [52]. A DNA fragment encoding Dam was amplified from the pUASTattB-LT3-NDam vector (a gift from Dr. Andrea H. Brand) and subsequently combined with GFP and Stil fragments via the In-Fusion cloning system (Takara). The resulting DNA fragments were cloned into the pUASp-K10-attB plasmid backbone [83]. These transgenic constructs were injected into embryos of attP-containing strains (Stil variants into P40, BDSC #25709, Dam and Dam-Stil into P2, BDSC #25710, respectively), and transformants expressing *mini-white* were selected. The sequence information of the primers used for transgenic fly generation is provided in S4 Table.

### Immunofluorescence staining

Immunofluorescence staining was performed as previously described [84]. Ovaries and testis from adults and the 3^rd instar larva were dissected in cold phosphate buffer saline (PBS) and fixed with PBS containing 5.3% (w/v) paraformaldehyde for 10 min. Fixed samples were washed in PBX [PBS with 0.2% (v/v) Triton-X (Nacalai)] for at least 30 min with several change of solution, then blocked with PBX containing 4% (w/v) bovine serum albumin (BSA) for at least 30 min at room temperature. After rinsing once with PBX, the samples were incubated with the primary antibodies diluted in PBX containing 0.4% (w/v) BSA for more than 4 h at room temperature or overnight at 4ºC. After several washes with PBX for at least 1 h, the samples were incubated in secondary antibodies diluted in PBX containing 0.4% (w/v) BSA for more than 2 h at room temperature. The samples were washed several times with PBX for at least 1 h and stained with 1 µg/mL of 4',6-diamidino-2-phenylindole (DAPI) (Sigma) diluted in PBX for 10 min at room temperature. Following two rinses with PBX and PBS, tissues were mounted in Fluoro-KEEPER Antifade Reagent (Nacalai). The following primary antibodies were used at the indicated dilutions: guinea pig anti-Vas (1:1000) [85], mouse anti-c-Myc [Wako, 9E10, 1:500], and rabbit anti-FLAG [MBL, PM020, 1:1000]. Secondary antibodies were Alexa Fluor 488-and 555-conjugated goat antibodies against rabbit, mouse and guinea pig IgG (Invitrogen, A11029, A11034, A11073, A21424, A21429 and A21435) at a 1:200 dilution in PBX containing 0.4% (w/v) BSA. Images were captured by Zeiss LSM780 Confocal Microscope with 10x, 20x, and 40x lens and processed with Zen (Zeiss) and Fiji application [86].

## TUNEL assay

TUNEL assay was conducted using ApopTag Kits (Millipore), labelling the free 3'-OH DNA termini in situ with fluorescein-nucleotide by terminal deoxynucleotidyl transferase (TdT). Dissected ovaries in cold PBS were fixed with PBS containing 5.3% (w/v) paraformaldehyde and rinsed 3 times with PBX. Ovaries were washed with PBX for 10 min twice and further washed with PBST [PBS with 0.1% (v/v) Tween 20 (Nacalai)] twice. After rinsing with Equilibration Buffer twice, the samples were incubated with TdT reagent [70% (v/v) Reaction Buffer and 30% (v/v) TdT Enzyme] for 1 h at 37ºC. After incubation with 1/35 diluted Stop/Wash Buffer for 15 min, samples were rinsed 3 times with PBX and then washed 3 times with PBX for 10 min. The tissue mounting and microscopic observation were as described in the "Immunofluorescence staining" section.

## RT-PCR and quantitative RT-PCR

Total RNA was isolated from ovaries using TRIzol LS Reagent (Invitrogen), following the manufacturer's protocol. RNA samples (1 µg each) were treated with DNase I (NEB) to digest genomic DNA, followed by inactivation of the enzyme with 5 mM EDTA at 75ºC for 10 min. Reverse transcription was performed using the DNase I-treated RNA with oligo(dT)20 primers and SuperScript III Reverse Transcriptase (Invitrogen), following the manufacturer's protocol. cDNAs were subjected to 30 cycles of RT-PCR with GoTaq DNA polymerase (Promega).

Quantitative PCR (qPCR) was performed with Fast SYBR Green regents (Invitrogen) using at least 3 biological replicates. Deltal-delta-CT method was used to obtain the relative transcript levels. *α-Tub84B (tubulin)* was used as the reference. The sequences of the primers are listed in S5 Table.

## mRNA-seq analysis

Total RNAs were extracted from 100 ovaries of control and *stil* mutant of *Tkv.CA OE* background and 40 ovaries of control and *stil* mutant of *P35 OE* background using TRIzol (ThermoFisher) following the manufacture's instruction. Extracted RNAs were treated with DNase I for 10 min at 37ºC and reactions were inhibited by 5 mM EDTA at 70ºC for 10 min, following that RNAs were purified using phenol/ chloroform/ isoamyl alcohol (mixed with a ratio of 25: 24: 1, pH 5.2). Purified RNAs were prepared for mRNA-seq libraries with NEBNext Ultra II RNA Library Prep Kit for Illumina (NEB), and deep-sequencing was conducted using NovaSeq6000 (Illumina). Sequencing analysis was conducted using bioinformatic software, CLC workbench 12 (QIAGEN). Adaptor sequences of total reads obtained from mRNA-seq were trimmed and reads were mapped to *Drosophila* genomes, r6.32 from Flybase (ftp://ftp.flybase.net/genomes/Drosophila_melanogaster/). After mapping the sequence data, Differential expression analysis was performed between control and *stil* mutant with *P35 OE* or *Tkv.CA OE* and calculated the expression level of each gene.

## DamID-seq and data processing

UASp-Dam and UASp-Dam-Stil flies were crossed with *NGT40; NosGal4-VP16* driver and maintained at 25°C. For each replicate, 100 ovaries of 50 flies were dissected in *Drosophila* Schneider's medium. Genomic DNA was extracted using the QIAamp Fast DNA Tissue Kit (QIAGEN cat. no. 51404), and methylated DNA was processed and amplified as described [51]. Briefly, genomic DNA was digested overnight with *Dpn*I (NEB), which cuts methylated GATC sites, and DamID adaptors were ligated to the resulting fragments. A subsequent digestion with *Dpn*II (NEB), which selectively cuts unmethylated GATC sites, was performed, and fragments containing consecutive methylated GATCs were then amplified by PCR using ligated adaptor-specific primers and PrimeSTAR HS DNA Polymerase (Takara).

The amplified DNA was sonicated using a Bioruptor (Diagenode) to obtain an average fragment size of 300 bp. DamID adaptors were removed by digestion with *Alw*I, and the DNA was purified using AMPure XP beads (Beckman Coulter). For

next-generation sequencing, the DNA was subjected to end-repair, A-tailing, Illumina adaptor ligation, and PCR amplification. Single-end 100 bp reads were subsequently generated on a NovaSeq 6000 platform (Illumina). Illumina NGS reads were aligned to the *Drosophila melanogaster* reference genome (r6.32), and enrichment profiles were generated using the damidseq_pipeline with default settings [87], with replicate libraries subsequently averaged. The Dam-Stil enrichment profile was normalized by the Dam control profile and Dam-Stil peaks were identified using find_peaks and annotated with the ChIPseeker pipeline [87].

### Fertility test

For female fertility assays, three virgin females were mated with three *y w* males for 3 days per replicate. For male fertility assays, three males were mated with three *y w* virgin females under the same conditions. After the mating period, eggs were collected for 24 hours, followed by incubation for an additional 24 hours. The total number of eggs and the number of hatched eggs were then counted for each replicate.

### Western blotting

5 pairs of ovary lysates mixed with 20% volume of 5×sample buffer [0.25% (w/v) Bromophenol blue, 0.5 M DTT, 50% (v/v) Glycerol, 10% (w/v) SDS, and 0.25 M Tris-HCl (pH 6.8)] were separated by SDS–PAGE and transferred onto 0.45 µm Clear Trans PVDF membranes (Wako). Membranes were blocked with 3% (w/v) skim milk (Nacalai Tesque) in PBST and incubated with primary antibodies diluted in HIKARI signal enhancer (Nacalai Tesque). The primary antibodies used were mouse anti-Myc [Wako, 9E10, 1:1000] and mouse anti-α-Tubulin [Santa Cruz, DM1A, 1:2000]. HRP-conjugated secondary antibodies used were goat anti-mouse IgG [Bio-Rad, 170–6516, 1:3000]. Chemiluminescence detection was performed using the Chemi-Lumi One reagent kit (Nacalai), and immunoreactive bands were visualized with a ChemiDoc Touch imaging system (Bio-Rad) and analyzed using Fiji [86].

### S2 cell culture experiments

*Drosophila* Schneider S2 cells were cultured at 28°C in Schneider's medium (Thermo) supplemented with 10% (v/v) fetal bovine serum (FBS), 100 U/mL penicillin, and 100 µg/mL streptomycin (Thermo). Plasmids encoding Stil variants were generated using the Gateway cloning system with (Life Technologies) with pAFW, which drives the expression of N-terminal FLAG-tagged fusion proteins under the control of the Actin5C promoter. *rpr*-GFP and *ubiP*-GFP reporter plasmid was constructed by replacing 2.4-kb tubulin promoter of the *tubP*-GFP, a gift from Dr, Steve Cohen [88], with the PCR-amplified 5-kb promoter region and 5'-UTR of *rpr* and the PCR-amplified 2,137 bp promoter region and 5'-UTR of *ubi-p63E*.

For the reporter assay, a total of 4 µg of plasmid DNA was used for each transfection. Transgenes and reporter plasmids were included at 1 µg each as needed, and the remaining amount was supplemented with the control plasmid pAW without attR sites to reach 4 µg. The plasmid mixture was combined with 20 µL of Hilymax transfection reagent (Dojindo) and transfected into 1 mL of S2 cells seeded one day prior, following the manufacturer's instructions. Two days of post-transfection, cells were collected, sonicated, and followed by centrifugation. The supernatant was mixed with 20% volume of 5×sample buffer. Proteins were then analyzed by Western blotting using the following antibodies and Signal Enhancer Reagent HIKARI (Nacalai): mouse anti-α-Tubulin [Santa Cruz, DM1A, 1:1000] rabbit anti-GFP [Clonetech, 632569, 1:1000], and rabbit anti-FLAG [MBL, PM020, 1:1000]. Chemiluminescence detection was performed using the Chemi-Lumi One reagent kit (Nacalai), and immunoreactive bands were visualized with a ChemiDoc Touch imaging system (Bio-Rad) and analyzed using Fiji [86].

For immunostaining, transfected S2 cells were collected at 2-days post-transfection and plated onto poly-L-Lysine-coated coverslips for at least 30 min to promote adhesion. Cells were fixed in 5.3% (w/v) paraformaldehyde for 15 min,

permeabilized with PBX for 10 min, and washed three times with PBX. After blocking with PBX containing 4% (w/v) BSA for 30 min, cells were incubated with primary and secondary antibodies for at least 1 hour each, followed by nuclear staining with 1 µg/mL of DAPI for 10 min. After rinsing with PBS, samples were mounted with Fluoro-KEEPER Antifade Reagent (Nacalai). Images were acquired using a Zeiss LSM780 and 900 confocal microscopes with a 63 × objective lens and processed using Zen and ImageJ software.

## Supporting information

**S1 Fig. *stil* function is indispensable in female germline cells not male germline cells.** (A) Immunostaining of ovaries from *y w*, *stil³/CyO*, *stil³/Df*, and *stil³/stilEY16156* with antibody against Vasa (red) with DAPI (blue). Scale bar: 50 µm. (B) Quantification of the percentage of ovarioles containing germline cells per ovary in 2–3-day-old females. Each dot represents an individual ovary. Genotypes are indicated below the graph and the number of germarium assessed is noted above each bar. Error bars represent standard deviation (s.d.). (C) Immunostaining of the apical end of testes from *stilEY16156/CyO* and *stilEY16156* with anti-Vasa antibody (red) and DAPI (blue) (top panels). Seminal vesicles (SV) harboring sperms are also stained with DAPI (blue) (bottom panels). Asterisk denotes the apex of testis. Scale bar: 50µm (apical end of testes, top) and 20 µm (SV, bottom). (D) The numbers of egg laying and hatching rate. Daily egg laying by three *y w* females mated with three males of the indicated genotypes are shown (*n*=3). The number of ovarioles assessed is noted above each bar. Error bar indicates s.d.
(JPG)

**S2 Fig. Germline knockdown of *stil*, but not somatic knockdown, specifically disrupts egg development.** (A) Analysis of egg laying and hatching rates. The number of laid eggs and their hatching rates are measured daily for three females of the indicated genotypes: control, *stil*-germline knockdown (*stil*-GLKD) driven by *NGT40; NosGal4-VP16*, and *stil*-somatic knockdown (*stil*-STKD) driven by *tj-Gal4*, each mated with three *y w* males (*n*=3). Error bars indicate standard deviation (s.d.). (B) RT-PCR analysis of *sxl* and the control, *α-tub* transcripts in ovaries and testes from control and *stil*-GLKD flies, respectively.
(JPG)

**S3 Fig. Egg development is disrupted at the mid-stage of oogenesis in *stil* mutants rescued by P35 or DIAP1.** Immunostaining of ovarioles from *y w*, *stilEY16156/CyO; P35 OE* (*NGT40; NosGal4-VP16 > P35*), *stilEY16156; P35 OE* and *stilEY16156; DIAP1 OE* (*NGT40; NosGal4-VP16 > DIAP1*) flies with antibody against Vasa (red) and DAPI (blue). The enlarged image highlights a degenerated egg chamber at the mid-stage of oogenesis in *stilEY16156; P35 OE*. Scale bar: 50 µm.
(JPG)

**S4 Fig. BED-type zinc finger of Stil and transcriptome analysis of germline-rescued *stil* mutant ovaries.** (A) A schematic presentation of Stil protein, highlighting the BED-type zinc finger (BED, green) and the nuclear localization signal (NLS, blue). A multiple sequence alignment of Stil with three other proteins containing BED-type zinc finger motif is shown. The BED-type zinc finger motif consists of two cysteine and histidine residues (red) and aromatic residues (yellow) in the N-terminal region. (B) Transcriptome analysis of germline-rescued *stil* mutant ovaries expressing P35. Differential expressed genes (DEGs) are obtained with DEseq2. Volcano plot displays the adjusted *p*-value (*p*-adj) versus $\log_2$-fold change ($\log_2$FC) of expression level in *stilEY16156; P35 OE* compared to that in *stilEY16156/CyO; P35 OE*. Significantly upregulated genes ($\log_2$FC > 1, *p*-adj < 0.01) and downregulated genes ($\log_2$FC <-1, *p*-adj < 0.01) are highlighted in red and blue, respectively, while non-DEGs are shown in black.
(JPG)

**S5 Fig. *rpr*, but not *hid*, plays a central role in *stil*-deficient female germline cell death.** (A-C) Quantitative RT-PCR are performed on ovaries from germline-rescued *stil* mutants, as well as ovaries and testes with germline-specific *stil* knockdown using distinct drivers. Expression levels are normalized to *atub* with *rp49* serving as an internal control. (A) Early-stage ovaries are isolated by dissecting stage 4–5 oocytes from *stil$^{EY16156}$/CyO; P35 OE* and *stil$^{EY16156}$; P35 OE* flies (B) *stil* knockdown in female germline cells is achieved using *NGT40-Gal4; NosGal4-VP16*, *Bam-Gal4*, and *mata-tub-Gal4* drivers. (C) *stil* knockdown in male germline cells is achieved using *NGT40-Gal4; NosGal4-VP16* and *Bam-Gal4* drivers. Error bars represent standard deviation (s.d.). (D) Immunostaining of *stil$^{EY16156}$; hid$^{05014}$/+* and *stil$^{EY16156}$; hid$^{05014}$/Df(3L)X14* ovaries with antibody against Vasa (red) with DAPI (blue). Scale bar: 50 μm. (E) Quantification of the percentage of ovarioles containing germline cells per ovary in 2–3-day-old females. Each dot represents an individual ovary. Genotypes are indicated below the graph and the number of germarium assessed is noted above each bar. Error bars represent standard deviation (s.d.).
(JPG)

**S6 Fig. Stil variants localization and control validation in S2 reporter assay.** (A) Western blot analysis of 6×Myc-tagged Stil variants (FL, NT, CT, and AAYA) driven by *NGT40-Gal4; NosGal4-VP16*, with *y w* as a control. Stil variants were detected with anti-Myc, and α-Tubulin (αTub) served as a loading control. Arrowheads indicate Stil variant proteins. The lower panel shows quantification of the Myc/αTub signal ratio normalized to FL. Error bars indicate standard deviation (s.d.) (*n* = 3). (B) Immunofluorescence images showing Myc-tagged Stil variants (green) and DNA (DAPI, blue) in stage 3~4 egg chambers of the ovaries. Insets show enlarged views of nurse cell nuclei. Scale bar: 20 μm (egg chamber) and 5 μm (nurse cell nuclei). (C) Immunostaining of S2 cells expressing FLAG-tagged Stil variants with an anti-FLAG (green) antibody and DAPI (blue). Scale bar: 2 μm (D) Schematic representation of the *ubiP*-GFP reporter construct, in which the 2,173 bp upstream region and 5'-UTR of the *ubi-p63E* gene is fused to GFP. (E) Reporter assay in S2 cells followed by western blot analysis to detect GFP expression. Co-transfection of the *ubiP*-GFP reporter and FLAG-tagged Stil-FL were performed in S2 cells. An arrowhead indicates Stil-FL protein. *α*-Tubulin (αTub) is used as a loading control. The lower panel quantifies the GFP/αTub signal ratio, normalized to that in S2 cells transfected with the reporter alone. Error bars indicate standard deviation (s.d.) (*n* = 3).
(JPG)

**S7 Fig. Conservation of Rpr and Stil within Diptera.** Homologs of *Drosophila melanogaster* Rpr and Stil were identified by BLASTp, aligned, and analyzed phylogenetically. Homologs are present across Dipteran lineages, with the genus *Drosophila* highlighted in blue. Branch lengths indicate the expected number of substitutions per site, as shown by the scale bar.
(JPG)

**S1 Table. Differentially expressed genes identified by RNA-seq in P35-expressing *stil* mutant ovaries.**
(CSV)

**S2 Table. Differentially expressed genes identified by RNA-seq in Tkv.CA-expressing *stil* mutant ovaries.**
(CSV)

**S3 Table. Integrated DamID-seq and RNA-seq dataset.**
(CSV)

**S4 Table. List of primers used for the construction of transgenes in this study.**
(CSV)

**S5 Table. List of primers used for RT-PCR and qRT-PCR in this study.**
(CSV)

**S1 Data. Numerical data underlying graphs.**
(XLSX)

## Acknowledgments

We thank Dr. Andrea H. Brand (The Gurdon Institute, University of Cambridge), Dr. Andreas Bergmann (UMass Chan Medical School, Massachusetts), and Dr. Dahua Chen (Yunnan University Institute of Biomedical Research, Kunming) for their generous gifts of the Dam CDS plasmid, the UASp-P35 transgenic fly line, and the UASp-Tkv.CA transgenic fly line, respectively. We acknowledge the Bloomington *Drosophila* Stock Centre and the Kyoto Stock Center for the fly stocks. Confocal images and Western blotting data were acquired with LSM780, LSM900 and ChemiDoc Touch, respectively, at FBS Core Facility in The University of Osaka. We acknowledge the NGS core facility at the Research Institute for Microbial Diseases of The University of Osaka, Annoroad Gene Technology Co., Ltd., and Hangzhou Veritas Genetics Medical Institute Co., Ltd. for the sequencing. We appreciate the insightful discussion and suggestions from all the members of KT's laboratory.

## Author contributions

**Conceptualization:** Masaya Matsui, Shinichi Kawaguchi, Toshie Kai.

**Data curation:** Masaya Matsui, Shinichi Kawaguchi.

**Formal analysis:** Masaya Matsui, Shinichi Kawaguchi.

**Funding acquisition:** Masaya Matsui, Toshie Kai.

**Investigation:** Masaya Matsui, Toshie Kai.

**Methodology:** Masaya Matsui, Shinichi Kawaguchi, Toshie Kai.

**Project administration:** Shinichi Kawaguchi, Toshie Kai.

**Resources:** Masaya Matsui, Shinichi Kawaguchi, Toshie Kai.

**Software:** Masaya Matsui, Shinichi Kawaguchi.

**Supervision:** Shinichi Kawaguchi, Toshie Kai.

**Validation:** Masaya Matsui.

**Visualization:** Masaya Matsui, Toshie Kai.

**Writing – original draft:** Masaya Matsui, Toshie Kai.

**Writing – review & editing:** Masaya Matsui, Shinichi Kawaguchi, Toshie Kai.

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
