## [Decision Letter · Decision Letter 0]

30 Jan 2026

Dear Dr Kai,

We are pleased to inform you that your manuscript entitled "Transcriptional Repression of reaper by Stand Still Ensures Female Germline Development in Drosophila" has been editorially accepted for publication in PLOS Genetics. Congratulations!

Before your submission can be formally accepted, please read reviewer #1 comments that provide excellent suggestions for improving the presentation and readability of your manuscript. This is important so please take care to make these edits.

Also, before your submission can be formally accepted and sent to production you will need to complete our formatting changes, which you will receive in a follow up email. Please be aware that it may take several days for you to receive this email; during this time no action is required by you. Please note: the accept date on your published article will reflect the date of this provisional acceptance, but your manuscript will not be scheduled for publication until the required changes have been made.

Yours sincerely,

Giovanni Bosco, Ph.D.

Section Editor

PLOS Genetics

Giovanni Bosco

Section Editor

PLOS Genetics

Aimée Dudley

Editor-in-Chief

PLOS Genetics

Anne Goriely

Editor-in-Chief

PLOS Genetics

BlueSky: @plos.bsky.social

Comments from the reviewers (if applicable):

Reviewer's Responses to Questions

**Comments to the Authors:**

Reviewer #1: The authors demonstrate that the germ cell loss phenotype of stand still mutants is due to de-repression of the reaper gene. They demonstrate that reaper is a direct target of stil and provide a thorough analysis of the phenotype. Importantly they demonstrate a complete suppression of the stil phenotype by removing reaper. They have fully addressed the previous reviews including evidence that these protein families are conserved through evolution. This manuscript provides an important contribution in understanding the regulation of apoptotic genes in the ovary. I have minor suggestions for improvement of the manuscript.

Suggestions:

1. Add a sentence about evolutionary conservation to the abstract.

2. The DAPI is a bit hard to see in some of the figures such as the panels in 3E and 4D. It could be brightened or replaced with cyan.

3. In Figure S3, the p35 egg chambers appear to be degenerating abnormally, and may be partially “undead” as previously described (Mazzalupo and Cooley 2006, Baum et al. 2007). Undead egg chambers would indicate that p35 is functional but the egg chambers are attempting to die for another reason. As UASp-p35 is normally robustly expressed with NGT;nosGal4 in mid-oogenesis, it seems unlikely that the expression level is insufficient, and likely something else is going on if oogenesis is not restored normally. Since oogenesis is not restored normally in p35 OE, the text on lines 446-7 should be clarified.

4. Could also cite Arya et al. 2015 on lines 265-7 who demonstrated that abdA regulates rpr expression. Interestingly abdA is differentially expressed in the data in Fig3C.

5. It appears that stil-AAYA rescues better than CT based on vasa expression and this should be acknowledged on p.14.

6. Should also consider citing Mehrotra et al. 2008 who demonstrated repression of rpr in endocycling cells and perhaps should be included in the discussion lines 466-7.

Reviewer #2: The authors have addressed almost all the points that were brought up. Two issues that need to be addressed. Figure 6B-C still needs a scale for the chromatin marks on the side. I don't think the data shows that rpr is a direct target of the chromatin mark based changes but I understand that this is the authors interpretation of the data.

**Have all data underlying the figures and results presented in the manuscript been provided?**

Reviewer #1: Yes

Reviewer #2: Yes

PLOS authors have the option to publish the peer review history of their article (what does this mean? ). If published, this will include your full peer review and any attached files.

**Do you want your identity to be public for this peer review?** For information about this choice, including consent withdrawal, please see our Privacy Policy .

Reviewer #1: No

Reviewer #2: No

**Data Deposition**

http://datadryad.org/submit?journalID=pgenetics&manu=PGENETICS-D-25-01273

**Press Queries**

---

## [Editor Report · Acceptance letter]

PGENETICS-D-25-01273

Transcriptional Repression of reaper by Stand Still Ensures Female Germline Development in Drosophila

Dear Dr Kai,

We are pleased to inform you that your manuscript entitled "Transcriptional Repression of reaper by Stand Still Ensures Female Germline Development in Drosophila" has been formally accepted for publication in PLOS Genetics! Your manuscript is now with our production department and you will be notified of the publication date in due course.

With kind regards,

Anita Estes

PLOS Genetics

On behalf of:
